# Rs11726196 Single-Nucleotide Polymorphism of the Transient Receptor Potential Canonical 3 (*TRPC3*) Gene Is Associated with Chronic Pain

**DOI:** 10.3390/ijms24021028

**Published:** 2023-01-05

**Authors:** Yoshinori Aoki, Daisuke Nishizawa, Seii Ohka, Shinya Kasai, Hideko Arita, Kazuo Hanaoka, Choku Yajima, Masako Iseki, Jitsu Kato, Setsuro Ogawa, Ayako Hiranuma, Junko Hasegawa, Kyoko Nakayama, Yuko Ebata, Tatsuya Ichinohe, Masakazu Hayashida, Ken-ichi Fukuda, Kazutaka Ikeda

**Affiliations:** 1Addictive Substance Project, Tokyo Metropolitan Institute of Medical Science, Tokyo 156-8506, Japan; 2Department of Dental Anesthesiology, Tokyo Dental College, Tokyo 101-0061, Japan; 3Department of Anesthesiology and Pain Relief Center, JR Tokyo General Hospital, Tokyo 151-8528, Japan; 4Department of Anesthesiology and Pain Medicine, Juntendo University Faculty of Medicine, Tokyo 113-8431, Japan; 5Department of Anesthesiology, Nihon University School of Medicine, Tokyo 173-8610, Japan; 6University Research Center, Nihon University, Tokyo 173-8610, Japan; 7Department of Surgery, Toho University Sakura Medical Center, Sakura 285-0841, Japan; 8Department of Anesthesiology, Saitama Medical University International Medical Center, Saitama 350-1298, Japan; 9Department of Oral Health and Clinical Science, Tokyo Dental College, Tokyo 101-0061, Japan

**Keywords:** *TRPC3* gene, SNP, chronic pain

## Abstract

Chronic pain is reportedly associated with the transient receptor potential canonical 3 (*TRPC3*) gene. The present study examined the genetic associations between the single-nucleotide polymorphisms (SNPs) of the *TRPC3* gene and chronic pain. The genomic samples from 194 patients underwent linkage disequilibrium (LD) analyses of 29 SNPs within and around the vicinity of the *TRPC3* gene. We examined the associations between the SNPs and the susceptibility to chronic pain by comparing the genotype distribution of 194 patients with 282 control subjects. All SNP genotype data were extracted from our previous whole-genome genotyping results. Twenty-nine SNPs were extracted, and a total of four LD blocks with 15 tag SNPs were observed within and around the *TRPC3* gene. We further analyzed the associations between these tag SNPs and chronic pain. The rs11726196 SNP genotype distribution of patients was significantly different from the control subjects even after multiple-testing correction with the number of SNPs. The TT + TG genotype of rs11726196 is often carried by chronic pain patients, suggesting a causal role for the T allele. These results contribute to our understanding of the genetic risk factors for chronic pain.

## 1. Introduction

Pain is the fifth vital sign, comprising an alert system that is necessary for human survival. However, prolonged pain causes a significant reduction in the quality of life. Chronic pain has deleterious effects on socioeconomic activity. In the United States, common pain conditions resulted in USD 60 billion (GBP 32.34 billion) in lost productivity each year, of which 77% was caused by lower performance rather than work absence [1]. In Australia, absent workdays are estimated to be 9.9 million work days annually, and the lower effectiveness of workdays is estimated to be 36.5 million work days, which raises productivity costs from AUD 1.4 billion (GBP 0.65 billion) to AUD 5.1 billion (GBP 2.35 billion) when absenteeism and presenteeism are considered [1]. In Japan, back pain (72.10%) and stiff shoulders (54.90%) were the most common types of pain. The respondents with these types of chronic pain reported more prolonged absenteeism (4.74% and 2.74%, respectively), incapacity for current employment (30.19% and 15.19%, respectively), overall work disability (31.70% and 16.82%, respectively), and indirect costs (JPY 1,488,385 and JPY 804,634, respectively) in 2016 [2]. The International Association for the Study of Pain revised the definition of pain in 2020 as “an unpleasant sensory and emotional experience that is associated with or similar to actual or potential tissue damage”, defining chronic pain as “pain that persists or recurs for more than 3 months” [3]. Chronic pain is associated with various diseases, and its incidence ranges from 8% to 60%. The International Statistical Classification of Diseases and Related Health Problems, 11th revision, was the first to include a classification code for chronic pain, classifying it according to condition and body site, but some chronic pain conditions, such as fibromyalgia and complex regional pain syndrome, have no known cause, and effective treatments and therapies have not been developed [3].

Various factors, including biological, physiological, and psychological factors, are involved in pain sensitivity and chronic pain, producing individual differences in chronic pain. The contribution of genetic factors to chronic pain has been reported by twin studies for each classification code. The contribution of the genetic factors to the incidence of chronic pain and severe chronic pain was reported to be 0.16 and 0.30, respectively [4]. Chronic pain has a strong genetic component. Inbred mice and rats exhibit varying pain sensitivities, and chronic pain runs in families in humans [5].

Several genetic polymorphisms have been reported to be involved in various chronic pain conditions. Fibromyalgia has been reported to be affected by single-nucleotide polymorphisms (SNPs) of such genes as *COMT*, *DRD4*, *MAO-A*, *GTPCH*, and genes that encode γ-aminobutyric acid-A (GABA_A_) receptors [6]. Chronic low back pain has been reported to be affected by SNPs of such genes as *OPRM1* and *COMT* [7]. A meta-analysis of all potential genetic variants that are associated with neuropathic pain identified 28 genes that were significantly associated with neuropathic pain, involving neurotransmission, the immune response, and metabolism. The genetic variants of the *HLA*, *COMT*, *OPRM1*, *TNFA*, *IL6*, and *GCH1* genes have been associated with neuropathic pain in multiple studies [7]. The genetic variants that were reported to be associated with chronic postoperative pain (postsurgical pain) include *COMT*, *GCH1*, *ABCB1*, *5HTR2A*, *IFNG1*, *IL1R1*, *IL1R2*, *IL4*, *IL10*, *IL13*, *NFKB1*, *HLA-DRB1*4*, *DQB1*, *PRKCA*, *CDH18*, *ATXN1*, *DRD2*, *NFKB1A*, *CHRNA6*, *KCND2*, *KCNJ3*, *KCNJ6*, *KCNK3*, *KCNK9*, *CACNG2*, *P2X7R*, *KCNS1*, and *TNFA* (Appendix A) [8]. Chronic widespread musculoskeletal pain has been reported to be related to the *RNF123* and *ATP2C1* loci [9]. The genetic polymorphisms of the exonuclease 3′–5′ domain containing 3 (*EXD3*) and solute carrier family 39 member 8 (*SLC39A3*) are associated with multiple chronic pain conditions [10]. A genome-wide association study meta-analysis of chronic low back pain reported associations between chronic low back pain and SNPs of the *SOX5*, *CCDC26/GSDMC*, and *DCC* genes [11]. A meta-analysis found the strongest association for the rs887797 SNP of the gene that encodes the protein kinase Cα (*PRKCA*) [12]. Genes that were significantly associated with multisite chronic pain in males, included *CENPW*, *MTCH2*, *NICN1*, *AMIGO3*, *DNAJA4*, *CTBP2,* and *NOP14*, and genes that were significantly associated with multisite chronic pain in females, included *NCAN*, *SPATS2L*, *TBC1D9*, *CAMK1D*, *SOX11*, *GON4L*, and *DAGLB* [13]. We also identified several SNPs that are possibly associated with chronic pain [14,15], in which several candidate SNPs, including SNPs within and around the *PRKCQ* and *HS3ST4* genes, were identified, and the most potent SNP was rs4773840 that was associated with postherpetic neuralgia (PHN). Various SNPs and genes have been reported to be associated with various types of chronic pain, but no clear picture has emerged for all genetic factors that are associated with chronic pain. This suggests that no single SNP or gene is responsible for chronic pain; instead, multiple SNPs and genes are involved.

The transient receptor potential (TRP) protein superfamily is composed of cation-permeable channels that are expressed in mammalian cells. As part of the oxidative stress response, there are three subfamilies of TRPs: the TRP cation channel subfamily C (TRPC; which contains calcium entry channels), the TRP cation channel subfamily M (TRPM; which is involved in biological proliferation and death), and the TRP cation channel subfamily V (TRPV; which is activated by chemical, mechanical, and physical stimuli) [16]. Among these, at least six channels (TRPV1-4, TRPM8, and TRPA1) are expressed in the nociceptors, where they act as transducers of signals from thermal, chemical, and mechanical stimuli and play crucial roles in the generation and development of pathological pain perception [17,18]. With regard to chronic pain, TRPV1, TRPV3, TRPM3, and TRPA1 are particularly interesting [19,20,21,22]. The antagonists of TRPV1, TRPV3, and TRPA1 have been advanced into clinical trials for the treatment of inflammatory, neuropathic, and visceral pain [20]. Additionally, TRPV4 and TRPM8 were reported to be involved in the mechanisms of chronic pain [23,24] and are suggested to be putative treatment targets for chronic low back pain [23]. For the TRPC channels, TRPC1, TRPC4, and TRPC5 are activated directly by inositol-1,4,5-trisphosphate, and TRPC3, TRPC6, and TRPC7 are activated by diacylglycerol (DAG), independent of the storage reduction of intracellular calcium [16]. The TRPC channels have been reported to play important roles in both the nociceptor signaling and the sensitization of the nociceptors by the inflammatory mediators [25]. This previous study [25] revealed a major contribution of TRPC to the neuronal calcium homeostasis in the somatosensory pathways. These channels are non selectively permeable to cations, with variable selectivity for calcium over sodium among the different members. Here, TRPC engages in both the store-operated calcium entry (SOCE) and the receptor-operated calcium entry (ROCE) in the control of the calcium influx that triggers the calcium-dependent pathways and the peripheral sensitization. Therefore, TRPC is functionally coupled to several inflammatory transduction mechanisms, including the UTP/P2Y2 and protease/PAR2 signaling complexes. This unique dual contribution to SOCE and ROCE defines the calcium-permeable TRPC, including TRPC3, as a key regulator of the calcium homeostasis in the dorsal root ganglia (DRGs) neurons under normal and pathological pain conditions. Xia et al. (2015) suggested that TRPC3 may be a novel therapeutic target for chronic pain [26]. We previously reported that a SNP near the *TRPC3* gene was associated with the postoperative fentanyl requirements and pain [27]. Several studies of the *TRPC3* gene polymorphisms and chronic itch have been reported [25,28], but no studies have evaluated the associations between the *TRPC3* gene polymorphisms and chronic pain. In the present study, we investigated the genetic polymorphisms within and around the *TRPC3* gene and analyzed the *TRPC3* gene polymorphisms that influence chronic pain.

## 2. Results

The sample groups in the present study were the neuropathic pain patient group (NPPG) and the nociceptive control group (NOCG), without any diseases or chronic pain, as detailed in a previous report [14]. The genotype data for the 50 kilobase pair (kbp) regions upstream and downstream of the *TRPC3* gene were extracted from the whole-genome genotyping data (Appendix A). The linkage disequilibrium (LD) blocks were made with HaploView for 29 SNPs in this region using the data from the NPPG. Among the 29 SNPs that were calculated by HaploView, three SNPs were excluded because they had a minor allele frequency of 0 (Table 1). The *r*^2^ value, a measure of the LD strength, was calculated. As a result, 15 SNPs were selected as tag SNPs because they had other SNPs with a greater LD between them (Table 1). The *TRPC3* gene has been shown to be located on the complementary strand side, according to the National Center for Biotechnology Information dbSNP database. The alleles in each SNP were displayed, based on the dbSNP annotation. A total of four LD blocks with 15 tag SNPs (rs1507994, rs1314213, rs11726196, rs2135976, rs11732666, rs17517624, rs3762839, rs884701, rs13127488, rs906496, rs1358229, rs6838639, rs6817255, rs11098653, and rs13130390) were observed in the region within and around the *TRPC3* gene (Figure 1, Table 1). Among the 15 tag SNPs, rs11732666 and rs6838639 showed a significant difference, with Hardy–Weinberg equilibrium *p* values of 0.0406 and 0.0449, respectively, but the other SNPs were under the Hardy–Weinberg equilibrium (Table 1).

We further analyzed the associations between these tag SNPs and the chronic pain status. Even after correcting for multiple testing, such as the Bonferroni adjustments with the number of SNPs (the significance level after the Bonferroni correction was *p* < 0.0033 [0.05/15]), the genotype distribution for the rs11726196 SNP in the patient subjects was significantly different from the control subjects in the genotypic model (Pearson *χ*^2^ test, *p* = 0.00089; Table 2). This result suggests that rs11726196 is significantly associated with chronic pain. No other SNPs were significantly associated with the chronic pain status (Table 2).

The ratios of the TT/TG/GG genotypes of rs11726196 were 0.018/0.284/0.699 in the NOCG and 0.073/0.361/0.565 in the NPPG (Table 2, Appendix A). The T allele or TT + TG genotype ratio was higher in the NPPG than in the NOCG, compared with the C allele or GG genotype. This result suggests that the T allele of rs11726196 is considered to be the causative allele in the risk for chronic pain.

To examine in more detail the association between the *TRPC3* SNPs and chronic pain, a haplotype-based test was performed for the selected neighboring tag SNPs. As shown in Appendix A, strong associations were found for the haplotypes including the rs11726196 SNP with chronic pain. Although the strongest associations were observed between the haplotypes that consisted of one or two SNPs, including the rs11726196 SNP and chronic pain (*χ*^2^ = 12.7800, *p* = 0.0003), the associations were compromised when three SNPs, including the rs11726196 SNP, were incorporated in the analysis (Appendix A), and the trend was similar when four or more SNPs, including the rs11726196 SNP, were incorporated in the analysis (data not shown).

## 3. Discussion

The data distribution showed that postherpetic neuralgia (PHN) was the most common type of pain (94 patients [48.4%]), followed by spinal stenosis (20 patients [9.8%]), chronic postoperative pain (12 patients [6.2%]), cervical spondylosis (11 patients [5.7%]), disk herniation (seven patients [3.6%]), and traumatic nerve injury (seven patients [3.6%]; Table 3). Kosson et al. (2019) reported that the types of pain were osteoarticular pain (146 patients [44.92%]), neuropathic pain (139 patients [42.77%]), headache (43 patients [13.23%]), and other types of pain (21 patients [6.46%]) [29]. Their findings were similar to our data, in which 48.4% of patients had neuropathic pain. Bone and joint pain, for which our data were categorized separately, were less common (10.4% for scoliosis, 5.7% for cervical disc herniation, 4.7% for lower back pain, 3.7% for disc herniation, and 1% each for buttock pain, groin pain, olecranon joint pain, knee pain, and thoracic pain; total of 32.2%). No noteworthy differences were found between the groups of patients who attended the pain clinics in any of the study groups.

The present results suggest that the T allele of rs11726196 is a causative allele in the risk for chronic pain. Although this SNP is located in the intron, the SNP may exist in a functional region, such as a region that is involved in the transcription regulation. We searched for the functional region (i.e., promoter and enhancer) in the 5 kbp regions upstream and downstream of the rs11726196 SNP in a public database. The enhancers existed in the 2.2 kbp region upstream of rs11726196 and the 1.3 kbp region downstream of rs11726196 (Appendix A) [30]. The expression quantitative trait loci (eQTLs), identified by GTEx, showed that the *TRPC3* mRNA expression depends on the rs11726196 genotypes in the peripheral nerve, such as the tibial nerve, with TT > GG, which is the same trend as most peripheral tissues (Appendix A) [31]. The tibial nerve projects to the DRGs that are located in the peripheral nervous system. The dorsal root ganglion neurons may express TRPC3 in a similar trend as the tibial nerve. In the DRG neurons, the T allele carriers may exhibit an upregulation of the TRPC3 expression, possibly in the peripheral trigeminal ganglion (TGG) as well, which increases the intracellular calcium levels, possibly exacerbating the nociception and chronic pain, which is consistent with the present results. This suggests that the high TRPC3 expression in the peripheral nerves or an increase in the T allele number was associated with the higher risk of chronic pain in the present study. In the thoracolumbar DRGs in rats, the prolonged elevations of intracellular calcium can cause neuronal hyperexcitability and hypersensitivity, leading to nociception and chronic pain [26].

According to the 1000 genomes study in the dbSNP database, the allele frequencies for the T allele (i.e., complementary A allele) of the rs11726196 SNP are 0.5265, 0.1845, 0.3350, 0.492, and 0.281 in African, East Asian, European, South Asian, and American populations, respectively (https://www.ncbi.nlm.nih.gov/snp/rs11726196; accessed on 25 November 2022). Although the T allele frequency of this SNP appears to be relatively low in the East Asian population, including Japan, compared with other populations, the SNP would not be too uncommon in the East Asian population to detect some associations with susceptibility to chronic pain. Meanwhile, considering the relatively low T allele frequency, there might be a possibility that the East Asian population is less susceptible to chronic pain than other populations, in terms of the rs11726196 SNP, if this SNP actually contributes to the susceptibility to chronic pain.

The TRPC3 channels, as well as other TRP channels (e.g., TRPV1, TRPV3, TRPV4, TRPA1, TRPM3, and TRPM8), possibly play an important role in nociception [17,18,19,20,21,22,23,24,26]. In humans, *TRPC3* mRNA is highly expressed in the brain, including the cervical spinal cord and heart [32]. In the cervical spinal cord, *TRPC3* mRNA is most highly expressed in the dorsal horn, which contains ascending sensory neuron pathways. A single cell type analysis revealed that excitatory neurons express *TRPC3* mRNA [33]. The highest *TRPC3* mRNA expression was found in TGGs and DRGs from C1 to S1 in mice (seven cervical segments, 13 thoracic segments, six lumbar segments, and one sacral segment) [34]. The mechanotransduction complex of TRPC3 and TRPC6 is thought to be present in the sensory nerves [35]. The TRPC3 overexpression activates the P2Y2 purinergic receptors and protease-activated receptor-2 (PAR2/F2RL1), increasing the intracellular calcium levels [26]. In rat thoracic/lumbar DRGs, prolonged elevations of intracellular calcium are commonly associated with neuronal hyperexcitability or hypersensitivity, contributing to nociception and chronic pain [26]. However, in humans, the T allele carriers may exhibit an upregulation of the TRPC3 expression in the DRG neurons, possibly in TGGs as well, which increases the intracellular calcium levels, possibly leading to the exacerbation of nociception and chronic pain, as mentioned above. The activation of the P2Y2 receptors and PAR2 strongly elevates the intracellular calcium levels via the overexpression of TRPC3 [36]. Furthermore, a strong association was found between DAG and the activation of TRPC3 through the overexpression of the inflammatory receptors [36]. Diacylglycerol mediates the calcium influx by activating both SOCE and ROCE. This suggests that TRPC3 binds SOCE/ROCE and DAG as an inflammatory metabotropic receptor and is involved in the peripheral nerve sensitization [36]. Altogether, these reports suggest important roles for TRPC3 in the mechanisms of inflammation and pain.

We previously reported associations between the rs1465040 SNP near the *TRPC3* gene and postoperative analgesic requirements after the sagittal split ramous osteotomy and open abdominal surgery [27]. The TT + TC genotypes of the rs1465040 SNP of the *TRPC3* gene contributes to an increase in the postoperative fentanyl requirements and exacerbates pain, compared with the CC genotype. However, further investigations are required to understand the functional contribution of this *TRPC3* SNP to the postoperative analgesic requirements. Similar to rs1465040, a strong LD was observed for rs6534331 (*r*^2^ = 0.96) and rs4833787 (*r*^2^ = 1.00) in the sagittal split ramous osteotomy patient samples, based on the genotype data that were obtained in our previous study [27]. However, when eQTLs of these three SNPs were examined in GTEx, no significant association with the *TRPC3* gene expression was found. Therefore, based on the GTEx database, it is unlikely that these SNPs affect individual differences in opioid analgesia and pain through alterations of the gene expression. TRPC3 is considered to be important in analgesia/pain, but future studies are required to elucidate the mechanisms by which the SNPs influence the channel function or expression to lead to individual differences in the susceptibility to chronic pain.

In the additional haplotype-based test that was performed to examine in more detail the association between the *TRPC3* SNPs and chronic pain for selected neighboring tag SNPs, the strongest associations were observed for the haplotypes that consisted of one or two SNPs, including the rs11726196 SNP and chronic pain (*χ*^2^ = 12.7800, *p* = 0.0003; Appendix A). However, the associations were compromised when three SNPs, including the rs11726196 SNP, were incorporated in the analysis (Appendix A), and the trend was similar when four or more SNPs, including the rs11726196 SNP, were incorporated in the analysis (data not shown). The results suggest that the rs11726196 SNP, and no other SNPs, solely contributes to the susceptibility to chronic pain among the 15 tag SNPs that were examined, and this SNP would be the most plausible causal SNP.

## 4. Materials and Methods

### 4.1. Participants

The present study used patient group recruitment and sample collection methods that were approved by the ethics committees of the respective universities and hospitals (see the institutional review board statement section below). Patients who were enrolled in the study were 194 adult patients who suffered from chronic pain or pain-related disorders, including neuropathic pain, such as PHN (NPPG), and who visited the JR Tokyo General Hospital (Tokyo, Japan), Juntendo University Hospital (Tokyo, Japan), or Nihon University Itabashi Hospital (Tokyo, Japan) for chronic pain treatment. The detailed demographic data for the subjects and their statistics of attributes (age, sex) are provided in Table 3 and in a previous report [14].

Enrolled in the study as controls were 282 adult (20–70 years old, 158 males and 124 females), apparently healthy volunteers (NOCG) without any diseases or chronic pain who lived in or near the Kanto area in Japan. The detailed demographic data for the subjects and their statistics are provided in a previous report [14].

### 4.2. Genotyping

Genotyping was performed using HumanOmni1-Quad v. 1.0 (total markers: 11, 34, 514) and HumanOmniExpress-12 v. 1.1 (total makers: 7, 19, 665) for the 194 participants in the NPPG and HumanOmniExpressExome-8 v. 1.2 (total makers: 9, 64, 193) for the 282 subjects in the NOCG, respectively [14]. The genotype data for the SNPs in the 50 kbp region upstream and downstream of the *TRPC3* gene were extracted from microarray data for the NPPG and NOCG.

### 4.3. Linkage Disequilibrium Block Analysis

Single-nucleotide polymorphisms within and around the *TRPC3* gene were subjected to the LD statistics using HaploView 4.2 [37]. Single-nucleotide polymorphisms with minor allele frequencies < 0.01 were excluded from the LD analysis. The linkage disequilibrium was examined between all pairs of biallelic loci to calculate the correlation coefficients (*D*’ and *r*^2^) and the logarithm of odds (LOD). *D*’ and *r*^2^ were used to measure the LD strength and absolute LD between the pairs of SNPs, respectively.

### 4.4. Statistical Analysis

The statistical tests for the individual SNPs were performed using SPSS v. 21 software (IBM Japan, Tokyo, Japan). The associations between the genotypic variances of SNPs and chronic pain were statistically evaluated using the *χ^2^* test. To ensure that the observed genotype frequencies were in accordance with the expected frequencies for the entire population, we tested for deviations from the Hardy–Weinberg equilibrium. Two SNPs, rs11732666 and rs6838639, were outliers (Table 1). Multiple-testing corrections were performed using Bonferroni’s method to adjust the *p* values that resulted from the association analyses. In all of the association analyses, the criterion for significance was set at the corrected *p* values (i.e., crude *p* < 0.0033 [0.05/15]), and the criterion for significance was set at a crude *p* < 0.05 in the Hardy–Weinberg equilibrium test.

gPLINK v. 2.050 and PLINK v. 1.07 (https://zzz.bwh.harvard.edu/plink/index.shtml; accessed on 28 July 2022) [38] were used for the haplotype-specific tests that involved several SNPs among the 15 selected tag SNPs, with the “--hap-window” option to specify all haplotypes in sliding windows of a fixed number of SNPs (shifting one SNP at a time).

## 5. Conclusions

The present findings improve our understanding of the genetic risk factors for chronic pain. More studies are necessary to elucidate the genetic mechanisms of the individual differences in chronic pain sensitivity. The rs11726196 SNP of the *TRPC3* gene may be useful for predicting chronic pain susceptibility.

## Figures and Tables

**Figure 1 ijms-24-01028-f001:**
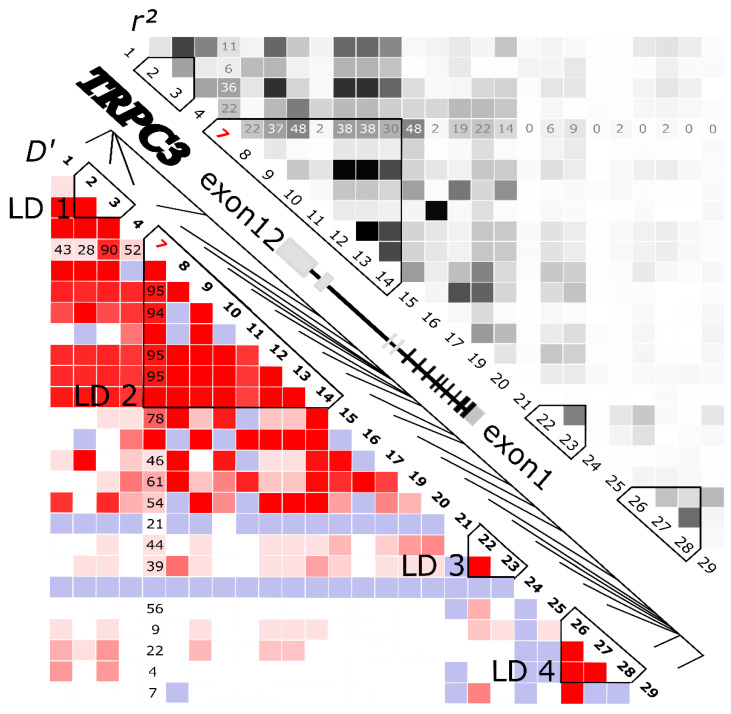
Schematic representation of the *TRPC3* gene structure and the LD. The gene structure of the *TRPC3* gene is shown with exons 1–12. The four linkage disequilibrium (LD 1–4) blocks were generated from 29 SNPs using the LD statistics and HaploView. Linkage disequilibrium was measured using the *D’* and *r*^2^ correlation coefficients and is displayed according to the standard color schemes for *D*’ (white: LOD < 2; *D*’ < 1; blue: LOD < 2; *D*’ = 1; shades of pink/red: LOD ≥ 2; *D*’ < 1; bright red: LOD ≥ 2; *D*’ = 1) and *r*^2^ (white: *r*^2^ = 0; shades of gray: 0 < *r*^2^ <1; black: *r*^2^ = 1). The *D*’ × 100 and *r*^2^ × 100 values of only the SNP pairs between rs11726196 (#7, red) and other SNPs are shown in each cell. Empty cells between these pairs of SNPs in the standard color scheme indicate *D*’ = 1. Red #7 indicates the tag SNP rs11726196.

**Table 1 ijms-24-01028-t001:** Results of the Hardy–Weinberg equilibrium test for each SNP.

LD Number	#	Tag	SNP	Position (hg19)	ObsHET	PredHET	HWpval	%Geno	FamTrio	MendErr	MAF	Rating	Results
	1		rs1048433	122748996	0.481	0.465	0.771	99	0	0	0.368		Used
LD1	2	Tag	rs1507994	122749541	0.382	0.346	0.2236	100	0	0	0.223		Used
LD1	3	Tag	rs13145213	122750079	0.527	0.493	0.4445	98.4	0	0	0.439		Used
	4		rs12499222	122774917	0.361	0.34	0.5514	100	0	0	0.217		Used
	5		rs6838198	122781263	0	0	0	0	0	0	0	BAD	-
	6		rs10518290	122805872	0	0	1	100	0	0	0	BAD	-
LD2	7	Tag	rs11726196	122806228	0.361	0.379	0.616	100	0	0	0.254		Used
LD2	8	Tag	rs2135976	122816181	0.131	0.131	1	100	0	0	0.071		Used
LD2	9	Tag	rs11732666	122824052	0.577	0.496	0.0406	99	0	0	0.458		Used
LD2	10	Tag	rs17517624	122824591	0.257	0.261	0.9592	100	0	0	0.154		Used
LD2	11	Tag	rs3762839	122825364	0.157	0.145	0.5814	100	0	0	0.079		Used
LD2	12	Tag	rs884701	122830406	0.56	0.495	0.0973	100	0	0	0.448		Used
LD2	13	Tag	rs13127488	122830800	0.56	0.495	0.0973	100	0	0	0.448		Used
LD2	14	Tag	rs906496	122833314	0.539	0.498	0.335	100	0	0	0.469		Used
	15		rs4292355	122842267	0.45	0.423	0.4946	100	0	0	0.304		Used
	16		rs12644087	122844819	0.152	0.14	0.6334	100	0	0	0.076		Used
	17		rs6841843	122857568	0.366	0.36	1	100	0	0	0.236		Used
	18		rs4833779	122861426	0	0	0	0	0	0	0	BAD	-
	19		rs950574	122864241	0.487	0.463	0.6025	100	0	0	0.364		Used
	20		rs970349	122871632	0.225	0.246	0.3395	100	0	0	0.144		Used
	21		rs4001038	122886240	0.026	0.026	1	100	0	0	0.013		Used
LD3	22	Tag	rs1358229	122896734	0.555	0.499	0.1734	100	0	0	0.482		Used
LD3	23	Tag	rs6838639	122899165	0.524	0.452	0.0449	100	0	0	0.346		Used
	24		rs13109219	122900529	0.005	0.005	1	100	0	0	0.003		Used
	25		rs9999992	122902084	0.314	0.325	0.7639	100	0	0	0.204		Used
LD4	26	Tag	rs6817255	122902923	0.455	0.489	0.399	100	0	0	0.427		Used
LD4	27	Tag	rs11098653	122903090	0.257	0.254	1	100	0	0	0.149		Used
LD4	28	Tag	rs13130390	122903206	0.173	0.166	1	100	0	0	0.092		Used
	29		rs11098654	122909415	0.33	0.282	0.0251	100	0	0	0.17		Used

# Matches the number in Figure 1. Tag SNPs are indicated with a circle. MAF, minor allele frequency (those with a value of 0 were excluded [=BAD], and finally 26 [=Used] in the results column were used); ObsHET, observed heterozygosity *p* value; PredHET, predicted heterozygosity *p* value; HWpval, Hardy–Weinberg equilibrium *p* value; %Geno, percentage of the non-missing genotypes for this marker; FamTrio, number of fully genotyped family trios for this marker (0 for datasets with unrelated individuals); MendErr, number of the observed Mendelian inheritance errors (0 for datasets with unrelated individuals).

**Table 2 ijms-24-01028-t002:** Associations between the tag SNPs in and around the *TRPC3* gene and chronic pain, comparing the genotype distributions between patients and control subjects.

# Corresponding to Figure 1	SNP	Genotype	Number of Participants		*χ* ^2^
			NPPG	NOCG	*p*
#2	rs1507994	GG	112	177	0.816
		GA	73	97	0.665
		AA	6	8	
#3	rs13145213	GG	33	46	0.888
		GA	99	135	0.642
		AA	56	93	
#7	rs11726196	TT	14	5	14.0559
		TG	69	80	0.00089 *
		GG	108	197	
#8	rs2135976	GG	165	259	4.584
		GA	25	23	0.101
		AA	1	0	
#9	rs11732666	AA	48	102	7.395
		AG	109	129	0.025
		GG	32	51	
#10	rs17517624	AA	5	3	4.471
		AG	49	55	0.107
		GG	137	224	
#11	rs3762839	TT	161	229	2.439
		TG	30	50	0.295
		GG	0	3	
#12	rs884701	TT	52	102	5.324
		TC	107	129	0.070
		CC	32	51	
#13	rs13127488	TT	32	50	5.098
		TG	107	130	0.078
		GG	52	102	
#14	rs906496	AA	38	74	3.123
		AG	103	132	0.210
		GG	50	76	
#22	rs1358229	AA	39	72	5.687
		AG	106	125	0.058
		GG	46	85	
#23	rs6838639	TT	16	33	4.109
		TC	100	122	0.128
		CC	75	127	
#26	rs6817255	AA	66	82	1.194
		AC	87	136	0.551
		CC	38	57	
#27	rs11098653	AA	4	4	0.705
		AG	49	80	0.703
		GG	138	197	
#28	rs13130390	CC	1	3	0.692
		CT	33	54	0.708
		TT	157	225	

# Matches the number in Figure 1. NPPG and NOCG indicate the neuropathic pain patient group and the nociceptive control subject group, respectively. Three × 2 *χ*^2^ tests were conducted to compare the neuropathic pain patients and control subjects among the genotypes in each SNP. * Statistically significant differences between the neuropathic pain patients and control subjects. The *p* values in the table are the crude values without the Bonferroni correction.

**Table 3 ijms-24-01028-t003:** Distribution of the main diagnostic names and number/ratio of patients.

ICD-11	Diagnosis	Number/Ratio of Patients		Age (Avg)/Range	Sex (Male/Female/Unknown)
MG30.5	PHN (post-herpetic neuralgia)	94	0.485	71.8/32–89	45/45/4
FA82	Spinal canal stenosis	20	0.103	65.2/22–75	8/12/0
MG30.21	Chronic postoperative pain	12	0.062	59.9/28–88	3/9/0
FA8Z	Cervical spondylosis	11	0.057	59.8/30–74	5/6/0
MG30.02	Low back pain	9	0.046	60.0/41–72	4/5/0
FB1Y	Herniated disc	7	0.036	56.1/37–75	2/5/0
MG30.20	Post-traumatic pain	7	0.036	56.1/33–70	5/2/0
VV12	Hip pain	2	0.010	40.0/30–56	0/2/0
MD81.12	Inguinal pain	2	0.010	61.5/61–62	0/2/0
ME82	Knee joint pain	2	0.010	41.0/39–43	1/1/0
4B20	Sarcoidosis	2	0.010	51.0/33–69	0/2/0
ME84.3	Sciatica	2	0.010	64.5/54–75	1/1/0
	Thalamic pain	2	0.010	65.0/63–67	1/1/0
8B82.0	Trigeminal neuralgia	2	0.010	73.0/64–82	2/0/0
MD81	Abdominal pain	1	0.005	63/63	0/1/0
9A05	Blepharospasm	1	0.005	57/57	0/1/0
MD30	Chest pain	1	0.005	29/29	1/0/0
MG30.03	Facial pain	1	0.005	56/56	0/1/0
MG30.01	Fibromyalgia	1	0.005	68/68	0/1/0
	Fibula neuralgia	1	0.005	60/60	0/1/0
VV12	Generalized pain	1	0.005	35/35	0/1/0
MG30.02	Shoulder pain	1	0.005	65/65	0/1/0
MG30.03	Temporomandibular joint pain	1	0.005	63/63	0/1/0
	others	11	0.057	54.5/30–71	5/5/1
	Total	194	1.000	66.1/22–89	83/106/5

ICD-11, International Statistical Classification of Diseases and Related Health Problems, 11th revision; avg, average.

## Data Availability

The data in this study are available in the Appendix A.

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
