# Peer review of "Rs11726196 Single-Nucleotide Polymorphism of the Transient Receptor Potential Canonical 3 (TRPC3) Gene Is Associated with Chronic Pain"

_ijms, 2023, doi:10.3390/ijms24021028_

Round 1
Reviewer 1 Report
In the current manuscript, the authors examined the genetic associations between SNPs of the TRPC3 gene and chronic pain. However, the authors should include TRPV1, TRPV4, and possibly other TRP channels in their introduction and discussion as they an important role in nociception. In addition to that I have the following comments:
1) The result section is short and many data are presented as supplementary tables. As the manuscript contains only one figure and one table, I would suggest to move most of the supplementary tables to the main body of the manuscript.
2) I would suggest to put the genetic variants (page 2, lines 82-85) in a table.
3) Page 2, lines 96-97: please some details about your previous findings associating SNPs to chronic pain.
Kindest regards,
Author Response
ijms-1986053
Response to Reviewer 1
In the current manuscript, the authors examined the genetic associations between SNPs of the TRPC3 gene and chronic pain. However, the authors should include TRPV1, TRPV4, and possibly other TRP channels in their introduction and discussion as they an important role in nociception. In addition to that I have the following comments:
Response: According to the reviewer’s suggestion, we included a description of TRPV1, TRPV4, and other TRP channels in the Introduction and Discussion sections because they play an important role in nociception. [Lines 109-120, 257-258]
1) The result section is short and many data are presented as supplementary tables. As the manuscript contains only one figure and one table, I would suggest to move most of the supplementary tables to the main body of the manuscript.
Response: According to the reviewer’s suggestion, we moved Supplementary Tables S2 and S3 to the main body of the manuscript as Tables 1 and 3, respectively. [Lines 183, 224; Table 1; Table 3]
2) I would suggest to put the genetic variants (page 2, lines 82-85) in a table.
Response: According to the reviewer’s suggestion, we put the genetic variants (page 2, lines 82-85) in a table and added Supplementary Table S1. [Supplementary Table S1]
3) Page 2, lines 96-97: please some details about your previous findings associating SNPs to chronic pain.
Response: According to the reviewer’s suggestion, we provided some details about our previous findings that associated SNPs with chronic pain. [Lines 96-100]

Reviewer 2 Report
The manuscript by Aoky et al describes the association of the rs11726196 SNP of the TRPC3 gene with chronic pain which may be useful as a genetic marker for predicting susceptibility of some chronic pain conditions. Thus, the manuscript is of general importance to a large audience. Furthermore, the results showed in this work open an important question: How is the expression of this gene regulated? The SNP identified: Could it upregulated the TRPC3 expression and which is the molecular mechanism for this regulation?
I consider that this manuscript is appropriate to be accepted for publication after the following suggestion:
-The authors may consider include additional information about the Patient and control attributes (age, sex).
-The authors may include in discussion section, some information about the transcriptional regulation of this gene.
Author Response
ijms-1986053
Response to Reviewer 2
The manuscript by Aoky et al describes the association of the rs11726196 SNP of the TRPC3 gene with chronic pain which may be useful as a genetic marker for predicting susceptibility of some chronic pain conditions. Thus, the manuscript is of general importance to a large audience. Furthermore, the results showed in this work open an important question: How is the expression of this gene regulated? The SNP identified: Could it upregulated the TRPC3 expression and which is the molecular mechanism for this regulation?
I consider that this manuscript is appropriate to be accepted for publication after the following suggestion:
-The authors may consider include additional information about the Patient and control attributes (age, sex).
Response: According to the reviewer’s suggestion, we included additional information about the patient and control attributes (age, sex) in the manuscript and Table 3 (previous Supplementary Table S3). [Lines 224, 314-316; Table 3]
-The authors may include in discussion section, some information about the transcriptional regulation of this gene.
Response: Some information about transcriptional regulation of the TRPC3 gene was already included in the Discussion section, Supplementary Figure S1, and Supplementary Table S4. Thus, we did not mention this information further:
“We searched for the functional region (i.e., promoter and enhancer) in the 5 kbp regions upstream and downstream of the rs11726196 SNP in a public database. Enhancers existed in the 2.2 kbp region upstream of rs11726196 and 1.3 kbp region downstream of rs11726196 (Supplementary Table S4) [30]. Expression quantitative trait loci (eQTLs), identified by GTEx, showed that TRPC3 mRNA expression depends on rs11726196 genotypes in the peripheral nerve, such as the tibial nerve, with TT > GG, which is the same trend as most peripheral tissues (Supplementary Figure S1) [31].” [Lines 229-236; Supplementary Figure S1; Supplementary Table S4]

Reviewer 3 Report
The manuscript refers to an analysis of several genetic polymorphisms in patients with chronic pain. Despite the important number of patients and the data in supplementary files, there are several that were not analyzed. Tabla 1 is for individual SNP; however, where is the haplotype analysis? The SNP rs11726196 refers to an intron uncommon in the Asian population, according to PubMed, but the patients are Japanese. The authors do not elaborate on the subject. Another important issue is the lack of information on the patient type, comorbidities, and treatment.
In chronic pain, there are also other genetic polymorphisms on glutamate receptors, KCNJ6 gene polymorphisms for example
The discussion and conclusions should be rewritten
The table requires a legend
Author Response
ijms-1986053
Response to Reviewer 3
The manuscript refers to an analysis of several genetic polymorphisms in patients with chronic pain. Despite the important number of patients and the data in supplementary files, there are several that were not analyzed. Tabla 1 is for individual SNP; however, where is the haplotype analysis? The SNP rs11726196 refers to an intron uncommon in the Asian population, according to PubMed, but the patients are Japanese. The authors do not elaborate on the subject. Another important issue is the lack of information on the patient type, comorbidities, and treatment.
Response: According to the reviewer’s suggestion, we conducted an additional haplotype analysis. Descriptions of this analysis were added in the Materials and Methods, Results, and Discussion sections in the manuscript and Supplementary Table S3. Additionally, according to the 1000 genomes study in the dbSNP database, the allele frequencies for the T allele (i.e., complementary A allele) of the rs11726196 SNP are 0.5265, 0.1845, 0.3350, 0.492, and 0.281 in African, East Asian, European, South Asian, and American populations, respectively (https://www.ncbi.nlm.nih.gov/snp/rs11726196; accessed November 25, 2022). Although the T allele frequency of this SNP appears to be relatively low in the East Asian population, including Japan, compared with other populations, the SNP would not be too uncommon in the East Asian population to detect some associations with the susceptibility to chronic pain. We elaborated on this subject in the Discussion section. Although information about patient type, comorbidities, and treatment is described in a previous report [14], we added some further information about patient characteristics in Table 3 (previous Supplementary Table S3). [Lines 142-144, 173-181, 246-256, 296-305, 349-352; Table 3, Supplementary Table S3]
In chronic pain, there are also other genetic polymorphisms on glutamate receptors, KCNJ6 gene polymorphisms for example
Response: Although KCNJ6 gene polymorphisms are already described in the manuscript, we added information about the rs2835925 SNP that is associated with chronic postsurgical pain in Supplementary Table S1. Genetic polymorphisms of glutamate receptors have also been investigated [8], but it seems that positive associations with chronic pain have not been observed. Therefore, we did not further describe such polymorphisms. [Line 85; Supplementary Table S1]
The discussion and conclusions should be rewritten
Response: According to the reviewer’s suggestion, we rewrote parts of the Discussion and Conclusions sections.
The table requires a legend
Response: According to the reviewer’s suggestion, we added a legend in Table 3 (previous Supplementary Table S3). [Line 224; Table 3]

Round 2
Reviewer 3 Report
The manuscript has been modified by the authors according to the suggestions.